# Impact of Different Aquatic Exercise Programs on Body Composition, Functional Fitness and Cognitive Function of Non-Institutionalized Elderly Adults: A Randomized Controlled Trial

**DOI:** 10.3390/ijerph18178963

**Published:** 2021-08-25

**Authors:** Carlos Farinha, Ana Maria Teixeira, João Serrano, Hélder Santos, Maria João Campos, Bárbara Oliveiros, Fernanda M. Silva, Márcio Cascante-Rusenhack, Paulo Luís, José Pedro Ferreira

**Affiliations:** 1University of Coimbra, Research Unit for Sport and Physical Activity, CIDAF, UID/PTD/04213/2019, 3040-248 Coimbra, Portugal; cmnfarinha@gmail.com (C.F.); ateixeira@fcdef.uc.pt (A.M.T.); mjcampos@fcdef.uc.pt (M.J.C.); geral.fernandasilva@gmail.com (F.M.S.); 2Sport, Health & Exercise Research Unit (SHERU), Polytechnic Institute of Castelo Branco, 6000-266 Castelo Branco, Portugal; j.serrano@ipcb.pt; 3Coimbra School of Health Technology—IPC, 3046-854 Coimbra, Portugal; heldersantos98@gmail.com; 4Faculty of Medicine, University of Coimbra, 3000-370 Coimbra, Portugal; boliveiros@fmed.uc.pt; 5School of Physical Education and Sports, University of Costa Rica (UCR), San José 11501-2060, Costa Rica; marciocascante@gmail.com; 6Municipality of Sertã, 6100-738 Sertã, Portugal; pauloluis@cm-serta.pt

**Keywords:** physical exercise, water-based exercise, aging, elderly

## Abstract

Aquatic physical exercise programs have become progressively more popular among elderly people. Some of the major physical exercise program disadvantages on land are minimized due to the specific properties of the aquatic environment. The purpose of the present randomized controlled study is to verify the effects of different aquatic physical exercise programs on body composition, functional fitness and cognitive function in non-institutionalized elderly people. For this study, 102 elderly individuals were randomly allocated into four different groups: AerG (*n* = 25, 71.44 ± 4.84 years); IntG (*n* = 28, 72.64 ± 5.22 years); ComG (*n* = 29, 71.90 ± 5.67 years) and CG (*n* = 20, 73.60 ± 5.25 years). Individuals from the groups AerG, IntG and ComG participated in three different aquatic physical exercise programs for a period of 28 weeks. The CG participants kept to their usual routines. All participants were evaluated for body composition, functional fitness and cognitive function at two time moments, i.e., pre- (M1) and post-intervention (M2). Significant differences for body composition were found between M1 and M2 for FM (*p* < 0.001), LBM (*p* < 0.001) and WCir (*p* < 0.01) in the AerG, for BMI (*p* < 0.05), FM (*p* < 0.05), LBM (*p* < 0.001) and LCir-R (*p* < 0.05) in the IntG, and for WGT (*p* < 0.01), FM (*p* < 0.05), LBM (*p* < 0.01), LCir-R (*p* < 0.05) and LCir-L (*p* < 0.01) in the ComG groups. For functional fitness, differences were found between M1 and M2 for 2m-ST (*p* < 0.000), 30s-CS (*p* < 0.000), 30s-AC (*p* < 0.05), HG-T-R (*p* < 0.000) and HG-T-L (*p* < 0.000) in the AerG, for 2m-ST (*p* < 0.05), BS-R (*p* < 0.05), 30s-CS (*p* < 0.000), 30s-AC(*p* < 0.01), HG-T-R (*p* < 0.000) and HG-T-L (*p* < 0.000) in the IntG, and for 30s-CS (*p* < 0.000), HG-T-R (*p* < 0.000) and HG-T-L (*p* < 0.000) in the ComG groups. The present study evidenced the beneficial effects of physical exercise in an aquatic environment on body composition, functional fitness and cognitive function in non-institutionalized elderly adults. The ComG water-based exercise program showed more beneficial effects in the improvement of body composition and cognitive function variables, while the IntG and AerG programs were more effective in the improvement of functional fitness.

## 1. Introduction

Regular physical activity is a recognized cost-effective intervention for public health and is associated with an ever-widening constellation of health, economic, and other benefits, playing an important role in the prevention and management of many major chronic conditions. Thus, the implementation of programs, practices, and policies to facilitate more physical activity and limit sedentary behavior could result in significant health improvements and other benefits, as well as reducing the burden and cost of chronic disease to healthcare systems [1]. In this respect, sedentary behavior is associated with negative changes in the neuromuscular systems of healthy older adults, resulting in a decrease in physical functioning [2]. Body composition, functional fitness and cognition are variables that undergo negative changes during aging, and these changes can lead to the development of cardiometabolic diseases [3].

Evidence reveals that physical exercise on land has been identified as an effective method for improving body composition [4] and functional capacity [5], and reducing the cognitive impairment associated with aging [6]. However, exercise in an aquatic environment has progressively gained popularity, particularly in the elderly population, as it minimizes or overcomes some existing disadvantages from land-based exercise programs due to specific water properties, such as buoyancy and water viscosity. Such properties help to reduce strain on the joints and offer additional support for balance and other problems associated with a lack of strength, as is very frequent in frailty, providing a safer and more protective environment for physical exercise. In addition, aquatic exercise programs seem to be at least as or even more effective as land-based exercise programs [7] for improving elderly people´s health and well-being. 

A recent intervention study [8] that tested the impact of two walking programs (land-walking versus water-walking), concluded that the water program can be considered safer and is a preferred activity for elderly people, providing benefits in body composition that are similar or higher than those achieved in land-based exercise programs. Results showed that older participants benefit more from the lower impact forces and decreased risk of falls associated with water-walking, without compromising improvements in their cardiorespiratory fitness. 

A meta-analysis performed by Waller et al. [3] analyzed the effects of water-based exercise programs on functional fitness in healthy older adults compared to physical exercise programs performed on land. The authors concluded that physical exercise in aquatic environments seems to be effective in the maintenance and improvement of functional fitness in healthy older people. Moreover, when compared to exercise on land, exercise in an aquatic environment seems to be at least as effective and can be used as an alternative type of exercise, when exercise on land is not feasible or desirable [3]. 

Regarding cognitive function, another intervention study [8] tested the effectiveness of an aquatic exercise program in a group of older women, aged between 60 and 80 years old, and reported that aquatic exercises can contribute to the maintenance and improvement of cognitive function. 

For all these reasons, aquatic exercise programs appear to be a viable alternative to exercise on land, and it is crucial to continue exploring the potential of this environment in order to clarify its benefits and the effects of different water-based exercise programs. Thus, the aim of this randomized controlled trial is twofold. Firstly, to perceive the impact of different water-based exercise programs (continuous aerobic, aerobic interval and combined) on body composition, functional fitness and cognitive function in non-institutionalized older people. Secondly, to perceive which of the three water-based exercise programs will be more effective in improving function in non-institutionalized older people. According to the existing literature [3,5,8,9,10,11] we predicted that differences would be found in the experimental groups, as opposed to the CG, and that the ComG program would be more effective than the AerG and IntG programs.

## 2. Materials and Methods

### 2.1. Study Design

A randomized controlled trial was conducted in the central region of Portugal, between October 2018 and July 2020. The study included non-institutionalized older adults participating in three different aquatic physical exercise programs for a period of 28 weeks. A complete study protocol was previously published in [12]. Participants were recruited in order to analyze the impact of three aquatic exercise programs (over 28 weeks), namely: continuous aerobic exercise, aerobic interval exercise and combined exercise. Data regarding body composition, functional fitness and cognition were collected at two specific moments, pre-intervention (M1) and post-intervention (M2).

This study conformed to the Declaration of Helsinki guidelines and the protocol was approved by the Faculty of Sports Science and Physical Education, University of Coimbra Ethics Committee (reference: CE/FCDEF-UC/00462019). All participants provided written informed consent prior to participation.

### 2.2. Participants and Sample Size

The size and statistical power of the sample were calculated using the G*Power software application [13]. The following parameters were considered: F test (ANOVA); effect size: 0.25; α-level: 0.05; statistical power: 0.95; number of groups: 4; number of measures: 2 (pre- and post-intervention); margin for possible losses and refusals: 30%. Therefore, the initial size of the total sample was estimated at 76 participants.

Initially, 174 individuals from the community were personally invited to participate in the study. After applying the inclusion and exclusion criteria, 152 individuals were randomized into four different groups: the continuous aerobic exercise group (AerG, *n* = 36), aerobic interval exercise group (IntG, *n* = 41), combined exercise group (ComG, *n* = 48), and control group (CG, *n* = 27). According to the experience of the research team and according to previous studies [6,7,14], the rate of dropping out from exercise programs in the elderly population is high, so we gathered more participants. An external researcher used a computer-generated list of random numbers to allocate participants to each group in a ratio of 1:1:1:1. Researchers were blinded to ensure group randomization.

The inclusion criteria were the following: (a) participants of both sexes; (b) 65 years of age or older; (c) non-institutionalized older people; (d) having the autonomy to move from their residence to the municipal swimming pool; (e) individuals who gave permission to take part in the study by signing the consent form; (f) individuals with medical authorization to practice physical exercise in an aquatic environment, in cases where they have some type of clinical condition or comorbidity. The exclusion criteria were the following: (a) individuals with clinically diagnosed pathologies that put their health and that of others at risk during the practice of physical exercise in an aquatic environment; (b) having severe cognitive impairment, that is, a score lower than 9 points in the MMSE or mental illness that has been clinically diagnosed; (c) participants who attended less than 50% of the physical exercise sessions; (d) participants who failed to complete all of the proposed assessment tests.

All three experimental groups (AerG, IntG and ComG) performed different types of water-based exercise simultaneously for 28 weeks. Participants from the control group (CG) were asked to maintain their normal daily activities, including not performing any type of systematic physical exercise during the same time period. 

Figure 1 shows the entire allocation process for the different groups. Fifty participants were excluded from the study due to the following reasons: personal reasons (*n* = 11); the participant attended less than 50% of the exercise program sessions (*n* = 14); the participant did not complete all assessment tests (*n* = 12); injury not related to the intervention program (*n* = 4); and disease (*n* = 9). 

### 2.3. Outcomes Measurements

#### 2.3.1. Clinical and Demographic Characteristics

Data regarding age (AG), marital status (MS), regular medication (RMED), allergies (ALL), disease (DIS), annual medical appointments (AMA), sleep quality (SQ), alcohol, tobacco and drugs consumption (ATD) were assessed at baseline (M1) using a specific questionnaire developed for this purpose.

#### 2.3.2. Anthropometry

Anthropometric measurements were conducted by two certified investigators by FCDEF-UC. The following parameters of anthropometry were evaluated: height (HGT), using a portable stadiometer, Seca Bodymeter^®^ (model 208, Hamburg, Germany) with a precision of 0.1 cm; weight (WGT); body mass index (BMI); visceral fat (VF); percentage of fat mass (FM) and muscle body mass (LBM) using a portable scale TANITA BC-601 with a precision of 0.1 cm with 0.1 kg accuracy; and waist circumference (WCir); arm circumference (ACir) and leg circumference (LCir), using a retractable fiberglass tape (model Hoechst mass-Rollfix^®^, Sulzbach, Germany) with an accuracy of 0.1 cm.

#### 2.3.3. Physical Function

Physical function was assessed using the senior fitness test, developed by Rikli and Jones [15] and validated for the Portuguese population [16]: the strength of the lower and upper limbs was assessed using the chair stand test (30 s-CS) and arm curl test (30 s-AC), respectively (repetitions/30 s); aerobic capacity was assessed using the two-minute step test (2m-ST) (number of steps); flexibility of lower and upper limbs was tested with the chair sit and reach test (CSR) and back scratch test (BS), respectively (centimeters); and agility and dynamic balance through the timed up and go test (TUG) (seconds). Hand strength was assessed through the handgrip test (HG-T) using the Jamar hand dynamometer (Lafayette Instrument Company, Lafayette, IN, USA) (kg).

#### 2.3.4. Cognitive Function

Cognitive function was assessed with the Portuguese version of the mini-mental state examination (MMSE) [17]. The MMSE evaluates the following cognitive areas: orientation, short-term memory, attention and calculation capacities, long-term memory, and language capabilities. The final score has a maximum of 30 points, and scores below 24 can be used as an aid in the assessment of dementia. The test was used as an instrument to create a cognitive profile with the following criteria [18]: severe cognitive impairment (scores between 1 and 9 points); moderate cognitive impairment (scores between 10 and 18 points); mild impairment (score between 19 and 24 points), normal cognitive profile (scores between 25 and 30 points).

### 2.4. Intervention Protocol

The exercise programs were implemented by sport sciences and fitness experts with specific training in water aerobics and were developed according to the exercise prescription guidelines recommended by the American College of Sport Medicine (ACSM) for the elderly [19].

All exercise program sessions had a duration of 45 min, twice a week, for 28 weeks and were performed in a water environment (the water level was between 0.80 and 1.20 m, with a temperature of approximately 32 °C), using the rhythm of the music as a tool to control the intensity level of exercise. Each session had a maximum limit of 16 participants, with a ratio of 1 exercise coach for every 16 participants. Water exercise sessions were organized into three different sections: initial, main and final parts. 

The initial part, or the warm-up, lasted between 10 and 15 min, at low intensity (30–40% max HR), and were the same in the three water exercise programs. During this initial part, it was intended that the participants would adapt to the aquatic environment, i.e., to the water temperature, and provide muscular and metabolic stimulations to prepare the body for the main part of the session. Thus, simple exercises in water were used, such as displacements and isolated movements, with a progressive increase in complexity and intensity throughout the initial part.

The main part for each one of the three water exercise program sessions had a duration of 20 to 30 min and all were characterized as follows:In the water-based continuous aerobic program, aerobic exercises were used continuously throughout the main part of the session (20 to 30 min), with a target intensity of 60 to 70% of the maximum heart rate, according to the recommendations of the ACSM for the elderly [19];In the water-based interval aerobic program, the main part of the session consisted of performing exercises with different intensities, such as: short duration exercises of 30 seconds, with an intensity of 70–80% of the maximal heart rate, followed by active recovery intervals of 1 minute, using exercises with an intensity of 60–70% of the maximal heart rate;In the water-based combined training program (continuous aerobic and muscular strength), the main part of the session was divided into two phases, with equal time periods. The first phase consisted of aerobic exercises on a continuous basis, with a target intensity of between 60 and 70% of the maximal heart rate. In the second phase, muscle-strengthening exercises were applied (water environment): 6 to 8 different exercises, with auxiliary equipment to create more resistance to movement (e.g., “spaghetti” and dumbbells), covering muscle strengthening from trunk (e.g., rowing, inverted crunches, etc.), upper limbs (e.g., elbow flexion and extension, shoulder rotation, etc.) and lower limbs (e.g., knee flexion and extension, leg abduction and adduction, etc.). Strengthening exercises were implemented using 2 to 3 sets of 12 to 16 repetitions at a moderate intensity (6–7) on the Borg scale.

The final part of the water exercise sessions lasted between 5 and 10 min and was the same for each of the three water exercise programs. This part consisted of two phases: return to calm, where relaxation exercises were applied to resume the participants’ heart rate to values close to the resting state, and stretching, where exercises using a greater range of motion were used to stretch the main muscle groups used throughout the sessions.

### 2.5. Monitoring the Intensity of the Physical Exercise 

For safety and intensity target control reasons, all participants were randomly using heart rate monitors (Polar, RS800CX, Finland) during the exercise sessions, in all three water exercise programs. Depending on the heart rate values obtained, adjustments were performed to maintain the target intensity established for each water exercise program. 

The intensity of the different water exercise programs was predicted indirectly using the Karvonen formula [20]:

Target heart rate = ((maximal heart rate − resting heart rate) × % intensity) + resting heart rate.

Additionally, and to calculate the maximal heart rate, the formula created by [21] for the elderly was used:

Maximal HR = 207 beats per minute − (0.7 × chronological age).

### 2.6. Statistical Analysis

The collected data was subject to descriptive statistical analysis, where values such as maximum, minimum, mean and standard deviation were calculated for each variable in each assessment moment. Afterward, data normality was tested by considering the response to three conditions: z-values from the Skewness and Kurtosis tests; *p*-values from the Shapiro–Wilk test; and visual inspection of generated histograms. All longitudinal comparisons were performed using complete case analysis. Parametric data were analyzed using Student’s *t*-test for independent samples to compare the different moments (M1 and M2), and an ANOVA of one factor and post hoc Tukey’s test to analyze the differences between groups. Nonparametric data was analyzed using the Wilcoxon test to compare the different moments (M1 and M2), and the Kruskal–Wallis and Bonferroni tests to analyze differences between groups. Statistical analysis was performed using the statistical package for the social sciences (SPSS) statistical software, version 27.0. The level of significance used was *p* ≤ 0.05. 

## 3. Results

The final 102 participants completed the entire selection process (AerG: *n* = 25, 71.44 ± 4.84 years, 80% female; IntG: *n* = 28, 72.64 ± 5.22 years, 89.3% female; ComG: *n* = 29, 71.90 ± 5.67 years, 75.9% female; CG: *n* = 20, 73.60 ± 5.25 years, 55% female). Despite there being no inclusion/exclusion criteria based on gender, the sample was unintentionally formed mostly of female participants, mainly as a result of the higher number of female elderly participants who usually attend community aquatic exercise programs, compared to males. No adverse event was reported during the intervention, showing that the practice of physical exercise in an aquatic environment is a safe modality for the elderly population. Individual characteristics for each group at baseline are presented in Table 1. 

Group characteristics were very similar between groups at baseline. as no significant statistical differences were found for AG, MMSE, BMI, RMED, MS, ALL, DIS and ATD variables among participants from the AerG, IntG, ComG and CG groups. Most participants were taking medication intended to control their cholesterol, blood pressure and diabetes values. Body composition results were analyzed by type of aquatic exercise program before and after the intervention, and are presented in Table 2.

Global body composition results revealed that no significant statistical differences were found between groups, before and after intervention (Table 2). As a result of intervention with aquatic exercise, an increase in WGT was found in all exercise groups, with significant statistical differences on ComG (*p* = 0.007; Δ = 1.2%), but not in the CG. A similar increase was found for BMI in all aquatic exercise groups, with significant statistical differences in IntG (*p* = 0.041; Δ = 3.4%), but not in the CG. 

FM variable results revealed a significant statistical reduction in all aquatic exercise groups as a result of the intervention AerG (*p* = 0.009; Δ = −4.6%), IntG (*p* = 0.016; Δ = −2.4%) and ComG (*p* = 0.031; Δ = −2.2%), but not in the CG. LBM variable showed a significant increase in all aquatic exercise groups AerG (*p* = 0.007; Δ = 2.6%), IntG (*p* = 0.001; Δ = 3.7%) and ComG (*p* = 0.002; Δ = 3.1%), but not in the CG. Anthropometric results also showed a significant statistical reduction for WCir in the AerG (*p* = 0.018; Δ = −3.2%), for LCir-R in the IntG (*p* = 0.019; Δ = −2.0%) and in the ComG (*p* = 0.025; Δ = 2.6%), and for LCir-L in the ComG (*p* = 0.008; Δ = −3.1%). 

For functional fitness variables (Table 3), there were no significant statistical differences between groups before intervention (*p* > 0.05). However, we found significant statistical differences for 2 m-ST (*p* = 0.05) between participants from AerG and IntG, after intervention, as well as for CSR-R (*p* = 0.006), CSR-L (*p* = 0.004) and 30 s-CS (*p* = 0.018) variables, between participants from the AerG and CG, after intervention. In all identified cases, the results were higher in the AerG. Considering the time effects due to the intervention (M1 and M2), we found significant statistical differences for 2 m-ST in AerG (*p* = 0.000; Δ = 16.8%) and in the IntG (*p* = 0.015; Δ = 10.8%). For the BS-R variable, there was a global trend showing a reduction in the performance level in all groups but was just statistically significant for IntG (*p* = 0.026; Δ = 15.1%). For the 30 s-CS variable, we found a significant average increase in all three aquatic exercise groups: AerG (*p* = 0.000; Δ = 20.0%), IntG (*p* = 0.000; Δ = 15.4%) and ComG (*p* = 0.000; Δ = 23.1%), and an average reduction in the CG. For the 30 s-AC variable, we found an increase in average for all groups, and it was statistically significant in the AerG group (*p* = 0.010; Δ = 9.5%) and IGnt (*p* = 0.005; Δ = 11.8%). For the HG-T variable, we found a statistically significant increase in average in the three aquatic exercise groups, for both the right hand—AerG (*p* = 0.000; Δ = 27.3%), IntG (*p* = 0.000; Δ = 23.8%) and ComG (*p* = 0.000; Δ = 23.8%), and the left hand—AerG (*p* = 0.000; Δ = 19.0%), IntG (*p* = 0.000; Δ = 20.0%) and ComG (*p* = 0.000; Δ = 23.8%), but not for the CG. 

Table 4 shows the cognitive results variation, analyzed according to the type of aquatic exercise program, before and after intervention.

Regarding the cognitive function variable, no significant statistical differences were found between groups before and after the intervention (see Table 4). However, after the intervention, we found an increase in the MMSE average scores in the three aquatic exercise groups (between M1 and M2), with significant statistical differences in the ComG group (*p* = 0.008; Δ = 3.8%), but not in the CG. 

## 4. Discussion

The main purpose of the present study was to assess the impact of different aquatic exercise programs on the body composition, functional fitness and cognitive function of non-institutionalized elderly people. A preliminary systematic search revealed the innovative characteristics of the present study, as no other study was identified as assessing similar variables in an aquatic environment with elderly participants. 

For body composition, global results revealed that significant statistical differences were found for most of the variables analyzed, as a result of the intervention with aquatic exercise (AerG, IntG and ComG), while no differences were found in the CG. A more detailed analysis of the results revealed significant statistical differences over time (from M1 to M2), i.e., between pre- and post-intervention time moments in the AerG for the FM, LBM and WCir variables, in IntG for the BMI, FM, LBM and LCIr-R variables, and in the ComG group for the WGT, FM, LBM, LCir-R and LCir-L variables. The variance of the mean was higher for the FM variable in the AerG (Δ = −4.6%), for the LBM variable in the IntG (Δ = 3.7%), and for the LCir-D variable in the ComG (Δ = −2.6%). The increase of the WGT in all aquatic exercise groups may be associated with the significant statistical increase of LBM found in all exercise groups. 

A study conducted by [8] that tested the effectiveness of two walking programs at moderate intensity (both aquatic and land environment) on the body composition of elderly participants showed that both programs had beneficial effects on body composition. Like our results, [8] revealed that exercise in an aquatic environment provided a significant reduction in abdominal fat. In spite of there being no significant statistical differences for VF in our study, Wcir was reduced in all aquatic exercise groups, and that reduction was statistically significant in the AerG group. Additionally, in the same study, significant differences were found for LBM in the aquatic walking group. Similar results were found in the present study for all three aquatic exercise groups, with the IntG group obtaining the highest mean variation from all analyzed groups. The study by [8] also found significant differences for the LBM variable in the aquatic walking group. Similar results were found in the three aquatic exercise groups in our study, with the IntG group obtaining the highest mean variation when compared with the other two groups. However, in the study by [8], no significant statistical differences were found for BMI. In our study, the BMI increased in all aquatic exercise groups, and differences were statistically significant in the IntG group.

Another study, conducted by [3], tested the effectiveness of a high-intensity aquatic resistance program in a group of postmenopausal women, before and after 4 months of intervention. Significant statistical differences in FM were reported. The present study also found significant statistical differences for FM in all aquatic exercise groups, with the AerG group reporting the highest mean value. 

Previous results show that physical aquatic exercise seems to have a beneficial impact on body composition. Regarding the hypothetical effects of different types of aquatic exercise programs on body composition, functional fitness and cognitive function, results show that they all seem to have a similar positive effect on the FM and BMI variables, but the ComG group also showed significant statistical differences in the LCir variables, providing additional evidence to confirm that combined aquatic exercise programs may have a more positive effect on body composition.

Global results regarding functional fitness showed significant statistical differences in all aquatic exercise groups (AerG, IntG and ComG), but not in the CG. A more comprehensive analysis of the results showed that significant statistical differences were found between AerG and IntG for the 2 m-ST variable, and between AerG and CG for CSR-R, CSR-L and 30 s-CS variables, as a result of the intervention with aquatic exercise. Additionally, significant statistical differences were also found in the AerG group for 2 m-ST, 30 s-CS, 30 s-AC, HG-T-R and HG-T-L variables, in the IntG for 2 m-ST, BS-D, 30 s-CS, 30 s-AC, HG-T-R and HG-T-L variables and in ComG for 30 s-CS, HG-T-R and HG-T-L variables before and after intervention (M1 versus M2). Delta (Δ) variation of the mean was higher for 2 m-ST (Δ = 16.8%) and HG-T-D (Δ = 27.3%) variables in the AerG, for 30 s-AC (Δ = 11.8%) variable in the IntG, and for 30 s-CS (Δ = 23.1%) and HG-T-L (Δ = 23.8%) variables in the ComG group.

A study run by [22] assessed functional fitness in 13 individuals before and after a 12-week aquatic exercise intervention program. Sessions consisted of warm-up exercises on land, walking in the water, aquatic exercises with different materials, and swimming. Results showed significant statistical differences in agility, balance and the muscle strength of the lower limbs. A similar study conducted by [14] analyzed the impact of a 3-month aquatic exercise program with music and found significant statistical differences in upper- and lower-limb muscular strength, flexibility, aerobic capacity agility and dynamic balance. A third study conducted by [23], based on a 24-week high-intensity jumping exercise program, showed improvements in agility, dynamic balance, and lower limb muscular strength. Furthermore, [5] developed a 14-week land aerobic chair-based exercise intervention program and found significant statistical differences for upper-limb muscular strength, agility and dynamic balance. In the same study, a strength-training program using elastic bands showed significant statistical differences for lower-limb strength as a result of the intervention.

As evidenced in the studies above, the results of the present study also showed significant statistical differences in the muscle strength of upper and lower limbs and in hand-grip strength in all three aquatic exercise groups. Regarding aerobic capacity, significant differences were also reported between AerG and IntG, with AerG showing a higher mean variation. On the other hand, no significant statistical differences were found regarding flexibility, agility and dynamic balance. Thus, the present results support the idea that aquatic exercise programs seem to have positive effects on functional fitness, mainly in variables related to upper- and lower-limb strength. In spite of there being no evidence found in the present study, it seems that aquatic exercise programs also tend to have a positive impact on agility and dynamic balance [24,25] in older people. Such results are in agreement with those presented by [5] when comparing elderly institutionalized women participating in a chair-based aerobic exercise walking program, versus a chair-based elastic-band muscle strength program.

Finally, and regarding cognitive function, we found a global trend for positive improvements in cognitive function (i.e., through MMSE instrument) as a result of intervention with aquatic exercise in all exercise groups, but not in the CG. A previous study by [26] tested the efficacy of a water-based exercise program on physical fitness improvement and cognitive function stimulation in adult healthy women. Results showed significant improvements in participants’ cognitive function and the mental health domain, regardless of the group in which they were initially included, providing evidence that water-based exercise is capable of enhancing cognitive function and quality of life through improvements in mental health in healthy adult women. Similar results were reported by [9] when testing the efficacy of a 16-week moderate-intensity aquatic exercise program on elderly women´s BDNF levels and cognitive function. Thus, most studies show that aquatic exercise programs are beneficial to cognitive function and that moderate-intensity aerobic exercise seems to have high benefits on elderly people´s cognition levels. 

However, the present study also tested the efficacy of a combined aquatic exercise program (aerobic and muscle strength) on elderly non-institutionalized participants. The positive effects of strength-training exercise programs on elderly women´s cognition have previously been reported by [6]. The authors found significant statistical differences in older women´s levels of cognition after their participation in a 28-week elastic-band strength-training intervention program, supporting the idea that strength training is also an effective type of exercise program to be used by groups with cognitive impairment. In our study, combined exercise (aerobic and muscular strength) was responsible for a significant increase in MMSE results (*p* = 0.008) in older participants from M1 to M2, revealing that the combination between the two types of exercise programs may be even more beneficial for improving cognitive function in elderly people.

As regards the limitations of the study, we refer to the fact that it used a simple randomization method, instead of using a blocked randomization method, which would guarantee a balance in the number of participants in each group, reducing sequence unpredictability. Another limitation was in measurements: inter-observer reliability was not measured; however, the evaluators jointly trained the measurers to have the same methods and both evaluators were the same at the evaluation moments (M1 and M2).

Further research is needed, using different types of aquatic exercise programs, to confirm this hypothesis, as the limited number of studies published in the literature regarding aquatic exercise in older people was a limitation when trying to discuss the present results and for clarifying which type of exercise program is more effective in the elderly adult population.

## 5. Conclusions

Results from the present study provided evidence for the beneficial effects of physical exercise in an aquatic environment on body composition, functional fitness (namely in lower- and upper-limb muscle strength, handgrip strength and aerobic capacity), and cognitive function in non-institutionalized older adults. The present study also revealed that different types of aquatic exercise programs may have different impacts on body composition, functional fitness and cognitive function. ComG was more effective in the improvement of body composition and cognitive function variables, while IntG and AerG were more effective in the improvement of functional fitness. 

## Figures and Tables

**Figure 1 ijerph-18-08963-f001:**
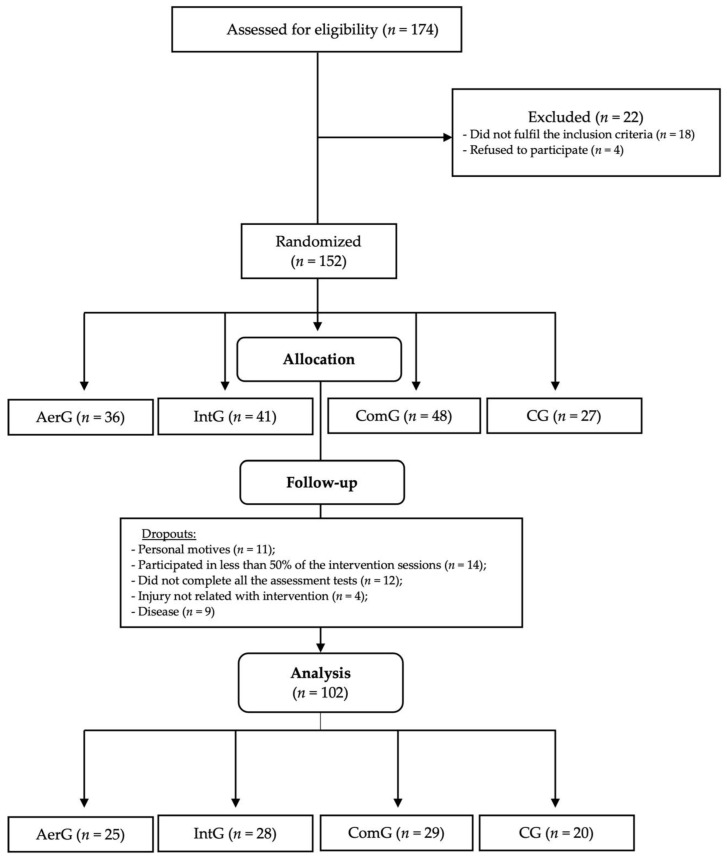
Allocation process for the different groups: continuous aerobic group (AerG); aerobic interval group (IntG); combined group (ComG); control group (CG).

**Table 1 ijerph-18-08963-t001:** Major sample characteristics at baseline.

Characteristics	AerG (*n* = 25)	IntG (*n* = 28)	ComG (*n* = 29)	CG (*n* = 20)	Value *p*
AG (years)	71.44 ± 4.84	72.64 ± 5.22	71.90 ± 5.67	73.60 ± 5.25	0.504
MMSE	27.00 ± 2.00	27.00 ± 2.00	26.00 ± 3.00	26.00 ± 3.00	0.489
BMI (kg/m²)	28.20 ± 3.30	29.10 ± 4.80	30.80 ± 5.30	29.50 ± 5.80	0.272
**RMED**	***n* (%)**	***n* (%)**	***n* (%)**	***n* (%)**	
Yes (%)	23 (92%)	27 (96.4%)	26 (89.7)	19 (95%)	0.768
No (%)	2 (8%)	1 (3.6)	3 (10.3%)	1 (5%)	
**MS**	***n* (%)**	***n* (%)**	***n* (%)**	***n* (%)**	
Married (%)	23 (92%)	20 (71.4%)	21 (72.4%)	19 (95.0%)	0.116
Divorced (%)	0 (0.0%)	2 (7.1%)	0 (0.0%)	0 (0.0%)	
Widow (%)	2 (8.0%)	6 (21.4%)	6 (20.7%)	1 (5.0%)	
Single (%)	0 (0.0%)	0 (0.0%)	2 (6.9%)	0 (0.0%)	
**ALL**	***n* (%)**	***n* (%)**	***n* (%)**	***n* (%)**	
Yes (%)	10 (40.0%)	9 (32.8%)	14 (48.3%)	7 (35.0%)	0.638
No (%)	15 (60.0%)	19 (67.9%)	15 (51.7%)	13 (65.0%)	
**DIS**	***n* (%)**	***n* (%)**	***n* (%)**	***n* (%)**	
No disease (%)	6 (24.0%)	5 (17.9%)	6 (20.7%)	3 (15.0%)	0.794
Cardiovascular (%)	4 (16.0%)	7 (25.0%)	4 (13.8%)	5 (25.0%)	
Metabolic (%)	6 (24.0%)	5 (17.9%)	5 (17.2%)	7 (35.0%)	
Cardiovascular and metabolic (%)	9 (36.0%)	11 (39.3%)	14 (48.3%)	5 (25.0%)	
**ATD**	***n* (%)**	***n* (%)**	***n* (%)**	***n* (%)**	
None (%)	22 (88.0%)	25 (89.3%)	26 (89.7%)	19 (95.0%)	0.772
Alcohol (%)	0 (0.0%)	0 (0.0%)	1 (3.4%)	0 (0.0%)	
Tabaco (%)	3 (12.0%)	3 (10.7%)	1 (3.4%)	1 (5.0%)	
Both (%)	0 (0.0%)	0 (0.0%)	1 (3.4%)	0 (0.0%)	

Note: Age (AG); mini mental state examination (MMSE); body mass index (BMI); regular medication (RMED); marital status (MS); allergies (ALL); diseases (DIS); alcohol, tobacco and drugs consumption (ATD); continuous aerobic group (AerG); aerobic interval group (IntG); combined group (ComG); control group (CG).

**Table 2 ijerph-18-08963-t002:** Body composition results variation analysed by type of aquatic exercise program before and after intervention.

	AerG			IntG			ComG			CG				
	M1 Mean (SD)	M2 Mean (SD)	Time (*p*)	M1 Mean (SD)	M2 Mean (SD)	Time (*p*)	M1 Mean (SD)	M2 Mean (SD)	Time (*p*)	M1 Mean (SD)	M2 Mean (SD)	Time (*p*)	Time × Group (M1)	Time × Group (M2)
WGT (kg)	70.5 (8.1)	71.0 (7.6)	0.150	71.3 (14.3)	71.9 (14.5)	0.097	75.1 (11.0)	76.0 (11.5)	0.007 *	75.5 (13.3)	75.2 (13.5)	0.589	0.334	0.350
BMI (kg/m²)	28.2 (3.3)	28.5 (3.0)	0.360	29.1 (4.8)	30.1 (6.2)	0.041 **	30.8 (5.3)	31.0 (5.1)	0.677	29.5 (5.8)	29.4 (5.9)	0.489	0.272	0.476
VF (%)	11.0 (3.0)	11.0 (3.0)	0.388	12.0 (3.0)	12.0 (3.0)	0.096	13.0 (3.0)	13.0 (3.0)	0.839	13.0 (6.0)	13.0 (6.0)	1.000	0.128	0.267
FM (%)	38.9 (7.3)	37.1 (8.4)	0.009 **	41.0 (6.7)	40.0 (6.7)	0.016 *	40.3 (9.8)	39.4 (9.8)	0.031 **	34.9 (10.9)	34.6 (10.7)	0.310	0.134	0.145
LBM (%)	26.5 (4.3)	27.2 (3.8)	0.007 **	24.5 (3.0)	25.4 (3.0)	0.001 **	25.5 (4.3)	26.3 (4.3)	0.002 **	27.7 (4.7)	27.9 (4.7)	0.188	0.079	0.160
WCir (cm)	99.7 (8.7)	96.5 (9.0)	0.018 *	102.2 (11.4)	101.1 (11.8)	0.216	105.1 (11.5)	103.4 (10.1)	0.109	103.1 (15.8)	102.5 (17.0)	0.652	0.409	0.181
ACir-R (cm)	31.5 (2.2)	31.5 (2.2)	0.729	32.7 (4.1)	32.2 (4.5)	0.107	32.7 (3.2)	32.4 (3.2)	0.331	31.7 (3.5)	31.6 (3.2)	0.745	0.659	0.734
ACir-L (cm)	31.6 (2.5)	31.3 (2.0)	0.381	32.3 (4.2)	31.7 (4.3)	0.850	32.4 (3.3)	32.2 (3.5)	0.475	31.6 (3.2)	31.4 (3.3)	0.278	0.861	0.616
LCir-R (cm)	54.0 (4.6)	53.5 (4.9)	0.443	54.9 (4.2)	53.8 (5.0)	0.019 *	54.6 (6.2)	53.2 (6.2)	0.025 *	52.5 (6.2)	51.8 (4.7)	0.247	0.093	0.375
LCir-L (cm)	53.7 (4.0)	53.1 (4.5)	0.266	53.8 (3.8)	53.3 (4.7)	0.265	54.3 (6.4)	52.6 (6.7)	0.008 *	51.0 (5.9)	50.6 (4.6)	0.466	0.153	0.312

Note: * *t*-Student test results; ** Wilcoxon test results; Correlation is significant at 0.05 level. Weight (WGT); body mass index (BMI); visceral fat (VF); fat mass (FM); muscular mass (LBM); waist circumference (WCir); right upper limb perimeter (ACir-R); left upper limb perimeter (ACir-L); right lower limb perimeter (LCir-R); left lower limb perimeter (LCIr-L); continuous aerobic group (AerG); aerobic interval group (IntG); combined group (ComG); control group (CG).

**Table 3 ijerph-18-08963-t003:** Functional fitness results variation analysed by type of aquatic exercise program before and after intervention.

	AerG			IntG			ComG			CG				
	M1 Mean (SD)	M2 Mean (SD)	Time (p)	M1 Mean (SD)	M2 Mean (SD)	Time (p)	M1 Mean (SD)	M2 Mean (SD)	Time (p)	M1 Mean (SD)	M2 Mean (SD)	Time (p)	Time × Group (M1)	Time × Group (M2)
2m-ST (no of steps)	80.9 (17.4)	94.5 (21.2)	0.000 **	71.5 (16.5)	79.2 (20.5)	0.015 *	81.6 (19.2)	85.4 (22.7)	0.250	74.3 (18.9)	79.4 (21.7)	0.147	0.069	0.048 †
CSR-R (cm)	−0.5 (6.6)	1.2 (8.7)	0.182	−3.7 (10.6)	−2.5 (10.5)	0.174	−3.5 (7.8)	−3.3 (8.1)	0.819	−7.6 (9.7)	−7.6 (7.0)	0.954	0.099	0.013 †
CSR-L (cm)	0.6 (7.2)	1.1 (8.8)	0.467	−3.9 (9.9)	−2.7 (11.6)	0.190	−5.8 (9.9)	−3.6 (8.6)	0.073	−3.5 (7.3)	−6.8 (8.2)	0.113	0.054	0.039 ††
BS-R (cm)	−9.9 (10.4)	−11.3 (11.4)	0.387	−11.9 (11.5)	−13.7 (12.8)	0.026 **	−14.3 (9.7)	−15.6 (9.7)	0.245	−16.6 (9.9)	−17.1 (10.2)	0.195	0.157	0.361
BS-L (cm)	−14.4 (7.2)	−15.9 (8.8)	0.098	−17.4 (8.6)	−18.1 (12.1)	0.631	−21.0 (10.8)	−21.5 (9.4)	0.369	−20.6 (10.7)	−21.2 (10.1)	0.197	0.056	0.241
TUG (s)	6.1 (1.1)	5.9 (0.9)	0.201	7.4 (1.8)	7.4 (2.2)	0.546	7.4 (3.0)	7.4 (2.9)	0.702	6.8 (1.7)	6.6 (1.9)	0.443	0.110	0.065
30s-CS (reps/30s)	15.0 (3.0)	18.0 (5.0)	0.000 *	13.0 (4.0)	15.0 (4.0)	0.000 *	13.0 (3.0)	16.0 (4.0)	0.000 **	15.0 (5.0)	14.0 (4.0)	0.229	0.185	0.016 †
30s-AC (reps/30s)	21.0 (6.0)	23.0 (6.0)	0.010 *	17.0 (7.0)	19.0 (6.0)	0.005 **	20.0 (5.0)	22.0 (6.0)	0.053	19.0 (6.0)	20.0 (6.0)	0.192	0.119	0.109
HG-T-R (kg)	22.0 (6.0)	28.0 (6.0)	0.000 **	21.0 (9.0)	26.0 (9.0)	0.000 **	21.0 (9.0)	26.0 (9.0)	0.000 **	24.0 (9.0)	25.0 (11.0)	0.259	0.411	0.317
HG-T-L (kg)	21.0 (6.0)	25.0 (6.0)	0.000 *	20.0 (9.0)	24.0 (9.0)	0.000 **	21.0 (9.0)	26.0 (8.0)	0.000 **	21.0 (10.0)	22.0 (10.0)	0.175	0.578	0.271

Note: * *t*-Student test results; ** Wilcoxon test results; † ANOVA and Turkey test results; †† Kruskal Wallis and Bonferroni test results. Correlation is significant at the 0.05 level. Two-minute step test (2m-ST); chair sit and reach test – right (CSR-R); chair sit and reach test – left (CSR-L); back scratch test – right (BS-R); back scratch test – left (BS-L); timed up and go test (TUG); chair stand test (30s-CS); arm curl test (30s-AC); hand grip test – rigth (HG-T-R); hand grip test – left (HG-T-L); continuous aerobic group (AerG); aerobic interval group (IntG); combined group (ComG); control group (CG).

**Table 4 ijerph-18-08963-t004:** Cognitive function results variation analysed by type of aquatic exercise program before and after intervention.

	AerG			IntG			ComG			CG				
	M1 Mean (SD)	M2 Mean (SD)	Time (*p*)	M1 Mean (SD)	M2 Mean (SD)	Time (*p*)	M1 Mean (SD)	M2 Mean (SD)	Time (*p*)	M1 Mean (SD)	M2 Mean (SD)	Time (*p*)	Time × Group (M1)	Time × Group (M2)
MMSE	27.0 (2.0)	28.0 (2.0)	0.294	27.0 (2.0)	28.0 (2.0)	0.138	26.0 (3.0)	27.0 (2.0)	0.008 **	26.0 (3.0)	26.0 (3.0)	0.651	0.489	0.255

Note: ** Wilcoxon test results. Correlation is significant at the 0.05 level. Mini mental state examination (MMSE); continuous aerobic group (AerG); aerobic interval group (IntG); combined group (ComG); control group (CG).

## Data Availability

All data are shown in the manuscript.

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
