# Peer review of "Impact of Different Aquatic Exercise Programs on Body Composition, Functional Fitness and Cognitive Function of Non-Institutionalized Elderly Adults: A Randomized Controlled Trial"

_ijerph, 2021, doi:10.3390/ijerph18178963_

Round 1

Reviewer 1 Report

This study added knowledge about the benefits of aquatic exercise on older adults. But I have some comments as follow.

  1. Please check the accuracy of the content, for example Line 68.
  2. Studies and review showed that aquatic exercise is beneficial to older adults on functional and cognitive fitness. The reasons to compare different aquatic exercises were unclear.
  3. For section 2.2, I suggested to present the sample size calculation first.
  4. Although it was claimed to be a ration of 1:1:1:1, the number of participants was imbalance in baseline. Please state the reasons.
  5. Please move Line 140-145 to Results section.
  6. Regarding waist circumference (WCir), arms 157 circumference (ACir) and legs circumference (LCir), who did the measurement? Did all the measurement measured by the same person? If not, how was the inter-rater reliability?
  7. Line 213, what is '1MR'?
  8. In the water combined training program, the description of second phase was different from 'water interval aerobic program'. The word 'combined' may cause misleading perception to readers as it was not the combination of continuous and interval aerobic program. 
  9. Line 221, all participants were randomly use heart rate monitors during the exercise. Please explain why you did not monitor all participants.
  10. Line 223 revealed that the interventions were delivered at both individual and group level. Please state clear which intervention was delivered at individual level, while which was at group level. Also, why the delivery methods were different between interventions?
  11. It was unclear how many sessions of interventions per week.
  12. As heart rate was monitored during the intervention, how many incidents were recorded as heart rate was over the intensity?
  13. Since there is a high drop-out rate (50/152), it is necessary to assess if it causes bias to the results. 

Author Response

My colleagues and I would like to thank you for the opportunity to resubmit our manuscript to the International Journal of Environmental Research and Public Health. We found that the reviewer’ comments were very helpful and we have done our best to incorporate all of their suggestions and reply to the reviewer` comments. We believe that this has made a significant contribution to the overall quality of the manuscript.

We send have included an updated version of our manuscript with all the changes highlighted in yellow. If you require any additional information, please do not hesitate to get in touch with us.

Point 1: Please check the accuracy of the content, for example Line 68

 Response 1: Thank you for your comment. We adjust the content as follows: “A meta-analysis performed by Waller et al. [3] analysed the effects of water-based exercise programs on functional fitness in healthy old adults compared to physical exercise programs performed in land. The authors concluded that physical exercise in aquatic environment seems to be effective in the maintenance and improvement of functional fitness in healthy old people. Moreover, when compared to exercise in land, exercise in an aquatic environment seems to be at least as effective and can be used as an alternative type of exercise, when exercise in land is not feasible or desired [3].” [line 68-74].

Point 2: Studies and review showed that aquatic exercise is beneficial to older adults on functional and cognitive fitness. The reasons to compare different aquatic exercises were unclear

Response 2:  Thank you for your comment. We adjust the content as follows: “For all these reasons, aquatic exercise programs appear to be a viable alternative to exercise on land, and it is crucial to continue exploring this environment in order to clarify its benefits and effects of different water-based exercise programs. Thus, the aim of this randomized controlled trial is twofold. First, to perceive the impact of different water-based exercise programs (continuous aerobic, aerobic interval and combined) on body composition, functional fitness and cognitive function in non-institutionalized old people. Secondly, to perceive which of the three water-based exercise programs will be more effective function in non-institutionalized old people. According to the existing literature [3, 5, 8, 9, 10, 11,] we believe that differences will be found in the experimental groups, as opposed to the CG, and that ComG program will be more effective than AerG and IntG programs.”. [line 79-89].

Point 3: For section 2.2, I suggested to present the sample size calculation first.

Response 3:  Thank you for your comment. Has been changed in the manuscript. [line 105-109].

Point 4: Although it was claimed to be a ration of 1:1:1:1, the number of participants was imbalance in baseline. Please state the reasons.

Response 4:  Thank you for your comment. Each aquatic exercise program was applied at different schedules. Before starting the practice, participants signed up at the most convenient schedules, but they only knew about the different types of programs later. So, there were schedules with more participants than others.

Point 5: Please move Line 140-145 to Results section.

Response 5:  Thank you for your comment. Has been changed in the manuscript. [line 246-251].

Point 6: Regarding waist circumference (WCir), arms 157 circumference (ACir) and legs circumference (LCir), who did the measurement? Did all the measurement measured by the same person? If not, how was the inter-rater reliability?

Response 6:  Thank you for your comment. Anthropometric measurements will be conducted by two a certified investigator by FCDEF-UC.

Point 7: Line 213, what is '1MR'?

Response 7:  Thank you for your comment. 1RM – One repetition maximim. Was wrong, but it was already been rectified. [line 209-210].

Point 8: In the water combined training program, the description of second phase was different from 'water interval aerobic program'. The word 'combined' may cause misleading perception to readers as it was not the combination of continuous and interval aerobic program. 

Response 8:  Thank you for your comment. We adjust the content as follows: “In the water combined training program (continuous aerobic and muscular strength), the main part of the session was divided into two phases, with equal time periods. The first phase was consisting of aerobic exercises on a continuous basis, with a target intensity between 60 and 70% of the maximal heart rate. In the second phase, muscle strengthening exercises were applied (water environment): 6 to 8 different exercises, with auxiliary equipment to create more resistance to movement (eg: spaghetti and dumbbells), covering muscle strengthening from trunk (eg: rowing, inverted crunsh, etc.), upper limbs (eg: elbow flexion and extension, shoulder rotation, etc.) and lower limbs (eg. knee flexion and extension, leg abduction and adduction, etc.).” [line 201-209].

Point 9: Line 221, all participants were randomly use heart rate monitors during the exercise. Please explain why you did not monitor all participants.

Response 9:  Thank you for your comment. The delivery of heart rate monitors to a certain number of participants in exercise sessions was because we did not have enough heart rate monitors for all participants.

Point 10: Line 223 revealed that the interventions were delivered at both individual and group level. Please state clear which intervention was delivered at individual level, while which was at group level. Also, why the delivery methods were different between interventions?

Response 10:  Thank you for your comment. All exercise interventions were performed in groups rather than individually. The delivery of heart rate monitors to a certain number of participants in exercise sessions was because we did not have enough heart rate monitors for all participants. The delivery of heart rate monitors was randomly carried out among the participants in each exercise session. In this way, we were able to monitor the subjects' heart rate, so that they were working at the desired intensity.

Point 11: It was unclear how many sessions of interventions per week.

Response 11:  Thank you for your comment. Twice a week. [line 179-180].

Point 12: As heart rate was monitored during the intervention, how many incidents were recorded as heart rate was over the intensity?

Response 12: Thank you for your comment. The participants during the sessions were sensitized to check if they were within the desired heart rate range. No incidents were verified.

Point 13: Since there is a high drop-out rate (50/152), it is necessary to assess if it causes bias to the results. 

Response 13: Thank you for your comment. We performed preliminary analyses to assess the risk of data bias. As mentioned in the statistical analysis section, we performed different analyzes to identify possible sources of bias, namely: z-values from Skewness and Kurtosis tests; p-value from Shapiro-Wilk test; and visual inspection of generated histograms. In general, the data does not present high sources of bias.

Reviewer 2 Report

First of all, thank you for the opportunity to review an article for possible publication in the prestigious International Journal of environmental research and public health.

The authors have developed an interesting article for the reader of IJERPH, which highlights the importance of exercise in the aquatic environment in elderly people. 

However, before its publication, I consider that there are some comments that should be taken into account:

1-. In the abstract, it is not necessary to present it in a structured way.

2-. Line 112. Was any scale on mobility or dependence on activities of daily living administered?

3-. Knowing that they are elderly people and that there may be an experimental death in the sample, do the authors give any explanation for the adherence to the different experimental programs proposed?

4-. Line 153-159. Adjust  interline spacing

5-. Line 156. How was visceral fat measured, was it a bioelectrical impedance? I looked up Seca's model and it doesn't look like an impedance to me. That is why it would be convenient to detail it

6-. Line 201. Add the ACSM reference from which it was taken.

7-. Due to the conclusions about the interest of the combined program, it would be convenient to further detail the strength exercise program (which exercises, how it has been controlled). It is not clear if it was performed in water, and if so, how was the intensity controlled?
8-. In line 201 it is described that 70% of the maximum heart rate is used and in lines 229-235 the use of the reserve heart rate is detailed. Clarify
In addition, exercise control in water, may require an adjustment in heart rate downward of about 10 beats per minute (Loupias, Jennifer Padilla M.S.; Golding, Lawrence A. Ph.D., FACSM Deep Water Running: A Conditioning Alternative, ACSM's Health & Fitness Journal: September-October 2004 - Volume 8 - Issue 5 - p 5-8). What is the author's opinion?

9-. Line 251 the IntG: has the same n as the AerG, correct it.

10-. It might be interesting to know the type of medication, since some exert a direct effect on the chronotropic level that should be considered.

11-. The authors have considered or controlled the feeding variable in some way, it would be interesting to detail it.

12-. The authors find an explanation as to why the AerG group has very significant improvements in upper limb tests if the upper limbs are not worked. It could be due to another type of activity that they will carry out. Perhaps it would be worthwhile to further detail the program carried out (what type of exercise) and explain it.

13-. As a conclusion, I think it is too broad to conclude that functional fitness is improved. It could be detailed which aspects are more disoriented, because although the variables that are improved are of great functionality, the TUG does not improve or the CSR.

14-. In the table legends it says "correlation is significant at the 0.05" I think the word correlation should be changed to avoid confusion with the statistical treatment of correlation.

15-. Revise the bibliographic references because they are not in the format required by the journal.

Author Response

My colleagues and I would like to thank you for the opportunity to resubmit our manuscript to the International Journal of Environmental Research and Public Health. We found that the reviewer’ comments were very helpful and we have done our best to incorporate all of their suggestions and reply to the reviewer` comments. We believe that this has made a significant contribution to the overall quality of the manuscript.

We send have included an updated version of our manuscript with all the changes highlighted in yellow. If you require any additional information, please do not hesitate to get in touch with us.

Point 1: In the abstract, it is not necessary to present it in a structured way.

Response 1: Thank you for your comment. Has been changed in the manuscript. [line 15-36].

Point 2: Line 112. Was any scale on mobility or dependence on activities of daily living administered?

Response 2: Thank you for your comment. No scale of mobility or dependence on activities of daily living was administered. Only elderly individuals from the community were selected.

Point 3: Knowing that they are elderly people and that there may be an experimental death in the sample, do the authors give any explanation for the adherence to the different experimental programs proposed?

Response 3:  Thank you for your comment. In the interior of Portugal, the aging rate is very high, and it is very common for this population to adhere to exercise programs in the aquatic environment.

Point 4: Line 153-159. Adjust interline spacing

Response 4:  Thank you for your comment. Has been changed in the manuscript [line 147-153].

Point 5: Line 156. How was visceral fat measured, was it a bioelectrical impedance? I looked up Seca's model and it doesn't look like an impedance to me. That is why it would be convenient to detail it.

Response 5: Thank you for your comment. The visceral fat was measured using the bioelectrical impedance method. We use was TANITA BC-601. It was wrong in the manuscript. It has already been corrected. [line 150].

Point 6: Line 201. Add the ACSM reference from which it was taken

Response 6:  Thank you for your comment. Has been changed in the manuscript [line 178].

Point 7: Due to the conclusions about the interest of the combined program, it would be convenient to further detail the strength exercise program (which exercises, how it has been controlled). It is not clear if it was performed in water, and if so, how was the intensity controlled?

Response 7:  Thank you for your comment. The exercises was performed in an aquatic environment, with auxiliary equipment to increase the resistance of the movements. The exercises was performed at a moderate intensity (6-7) on the Borg scale. [line 205-210].

Point 8: In line 201 it is described that 70% of the maximum heart rate is used and in lines 229-235 the use of the reserve heart rate is detailed. Clarify
In addition, exercise control in water, may require an adjustment in heart rate downward of about 10 beats per minute (Loupias, Jennifer Padilla M.S.; Golding, Lawrence A. Ph.D., FACSM Deep Water Running: A Conditioning Alternative, ACSM's Health & Fitness Journal: September-October 2004 - Volume 8 - Issue 5 - p 5-8). What is the author's opinion?

Response 8:  Thank you for your comment. This adjustment was not considered in the present study. However, participants were sensitized to check on their polar monitor if the heart rate value was within the intended range, and whenever there was any kind of discomfort or excessive tiredness, the participants were encouraged to alert and stop the exercise.

Point 9: Line 251 the IntG: has the same n as the AerG, correct it.

Response 9:  Thank you for your comment. Has been changed in the manuscript. [line 246-248].

Point 10: It might be interesting to know the type of medication, since some exert a direct effect on the chronotropic level that should be considered.

Response 10:  Thank you for your comment. The typology of regular medication was considered in this study. In most participants, the medication was intended to control cholesterol, blood pressure and diabetes values. [line 257-258].

Point 11: The authors have considered or controlled the feeding variable in some way, it would be interesting to detail it.

Response 11:  Thank you for your comment. There was no control on feeding. Participants were only asked about the consumption of alcohol, tobacco and drugs.

Point 12: The authors find an explanation as to why the AerG group has very significant improvements in upper limb tests if the upper limbs are not worked. It could be due to another type of activity that they will carry out. Perhaps it would be worthwhile to further detail the program carried out (what type of exercise) and explain it.

Response 12: Thank you for your comment. All exercise, regardless of the program, were performed in water environment, so whatever the movement there was always resistance caused by the specific properties of the aquatic environment. Thus, we believe that the significant increase in the strength of the upper limbs in the AerG comes from the practice of exercise in an aquatic environment.

Point 13:  As a conclusion, I think it is too broad to conclude that functional fitness is improved. It could be detailed which aspects are more disoriented, because although the variables that are improved are of great functionality, the TUG does not improve or the CSR.

Response 13: Thank you for your comment. We adjust the content as follows: “Results from the present study provided evidence for the beneficial effects of physical exercise in aquatic environment on body composition, functional fitness (namely in lower and upper limb muscle strength, handgrip strength and aerobic capacity), and cognitive function in non-institutionalized old adults.” [line 413-416].

Point 14:  In the table legends it says "correlation is significant at the 0.05" I think the word correlation should be changed to avoid confusion with the statistical treatment of correlation.

Response 14: Thank you for your comment. Has been changed in the tables 2, 3 and 4.

Point 15:  Revise the bibliographic references because they are not in the format required by the journal.

Response 15: Thank you for your comment. Has been changed in the manuscript. [line 431-495].

Round 2

Reviewer 1 Report

The authors do well to address my concerns, but some of them may need further clarifications.

  1. Line 109 showed the required total sample was estimated 76 participants. It was unclear why 152 participants were recruited. 
  2. "Each aquatic exercise program was applied at different schedules. Before starting the practice, participants signed up at the most convenient schedules, but they only knew about the different types of programs later. So, there were schedules with more participants than others." I am confusing because the number of participants was imbalance in group allocations. The different time to start the program or the convenient time of the participants would influence the attendance of the program, not the group allocation. Please describe the reasons of group imbalance.
  3. "Anthropometric measurements will be conducted by two certified investigators." It is necessary to show it to the readers too. Further on it, how's the inter-rater reliability among the assessors (e.g. kappa)?
  4. As all the interventions were delivered in group, please describe the group size and the ratio of participants and coach for future study. 
  5. As no adverse event was reported during the intervention, it may be worth to describe the safety of the interventions.
  6. Line 235-237, "data normality was tested by considering the response to three conditions: z-values from Skewness and Kurtosis tests; p-value from Shapiro-Wilk test; and visual inspection of generated histograms." This may not help to detect bias caused by drop-out. In turn, you may use other methods, such as last observation carried forward, to see if current results are changed. Please state what methods you used and describe the findings in your manuscript.

Author Response

My colleagues and I would like to thank you for the opportunity to resubmit our manuscript to the International Journal of Environmental Research and Public Health. We found that the reviewer’ comments were very helpful and we have done our best to incorporate all of their suggestions and reply to the reviewer` comments. We believe that this has made a significant contribution to the overall quality of the manuscript.

We send have included an updated version of our manuscript with all the changes highlighted in yellow. If you require any additional information, please do not hesitate to get in touch with us.

Point 1: Line 109 showed the required total sample was estimated 76 participants. It was unclear why 152 participants were recruited. 

Response 1: Thank you for your comment. We enroll 50% more persons in case of dropouts. According to the experience of the research team and according to previous studies (Chupel et al., 2017; Dziubek et al., 2015; Moreira et al., 2020), the rate of dropout from exercise programs in the elderly population is high, so we gather more participants. [line 114-116].

Point 2: "Each aquatic exercise program was applied at different schedules. Before starting the practice, participants signed up at the most convenient schedules, but they only knew about the different types of programs later. So, there were schedules with more participants than others." I am confusing because the number of participants was imbalance in group allocations. The different time to start the program or the convenient time of the participants would influence the attendance of the program, not the group allocation. Please describe the reasons of group imbalance.

Response 2:  Thank you for your comment. The simples randomization method was used which caused an imbalance in the number of participants in each group. The blocked randomization method should have been used. This aspect was mentioned as a limitation of the study: “As limitations of the study, we refer the fact that it was used the simple randomization method, instead of using the blocked randomization method, which would guarantee the balance in the number of participants in each group, reducing sequence unpredictability”. [line 415-418].

Point 3: "Anthropometric measurements will be conducted by two certified investigators." It is necessary to show it to the readers too. Further on it, how's the inter-rater reliability among the assessors (e.g. kappa)?

Response 3:  Thank you for your comment. This information has been added to the manuscript: “Anthropometric measurements will be conducted by two certified investigators by FCDEF-UC”. [line 149-150]. Inter-observer reliability was not measured. This information was added to limitations of the study: “Another limitation was in measurements inter-observer reliability was not measured however the evaluators jointly trained the measurements to have the same methods and both evaluators were the same at the evaluation moments (M1 and M2)”. [line 418-420].

Point 4: As all the interventions were delivered in group, please describe the group size and the ratio of participants and coach for future study. 

Response 4:  Thank you for your comment. Each session had a maximum limit of 16 participants with a ratio of 1 exercise coach for every 16 participants. [line 185-186].

Point 5: As no adverse event was reported during the intervention, it may be worth to describe the safety of the interventions.

Response 5:  Thank you for your comment. This information has been added to the manuscript: “No adverse event was reported during the intervention showing that the practice of physical exercise in an aquatic environment is a safe modality for the elderly population.” [line 256-258].

Point 6: Line 235-237, "data normality was tested by considering the response to three conditions: z-values from Skewness and Kurtosis tests; p-value from Shapiro-Wilk test; and visual inspection of generated histograms." This may not help to detect bias caused by drop-out. In turn, you may use other methods, such as last observation carried forward, to see if current results are changed. Please state what methods you used and describe the findings in your manuscript.

Response 6:  Thank you for your comment. We also have the same concerns about the drop-outs. However, we are not sure whether the reviewer is stating only one or two questions, and we will try to answer as detailed as possible. About data normality and longitudinal comparisons, they were performed with complete case analysis and, therefore, data adjustment to Gaussian distribution was assessed using the Shapiro-Wilk test using only subjects which have completed the study. As some theoretical distributions are robust to small deviations from normality when symmetrical and mesocortical, z scores from skewness and kurtosis were also computed.SPSS automatically generates histograms, PP ad QQ plots when describing quantitative data and thus those plots were also inspected. We understand the bias caused by drop-outs but we have prefered to use complete case analysis instead of imputing data, which was not missing completely at random (NOT MCAR) in many situations. By my experience, applying LOCF (last observation carried forward) to impute data may be dangerous and introduce more bias than complete case analysis, especially if data is not MCAR. For the sake of reproducibility and transparency, we will introduce the sentence: “All longitudinal comparisons were performed using complete case analysis." [line 241-242].

Round 3

Reviewer 1 Report

The revised manuscript addressed all my concerns and I have no further comments.